# Changing use of antidiabetic drugs in the UK: trends in prescribing 2000–2017

Samantha Wilkinson,[1] Ian Douglas,[1] Heide Stirnadel-Farrant,[2] Damian Fogarty,[3] Ana Pokrajac,[4] Liam Smeeth,[1] Laurie Tomlinson[1]

[1]Department of Non-Communicable Disease Epidemiology, London School of Hygiene and Tropical Medicine, London, UK
[2]GSK, Stevenage, UK
[3]Belfast Health and Social Care Trust, Belfast, UK
[4]West Herts Hospitals NHS Trust, Watford, UK

**Correspondence to**
Samantha Wilkinson;
Samantha.wilkinson@lshtm.ac.uk

## ABSTRACT

**Objectives** Guidelines for the use of drugs for type 2 diabetes mellitus (T2DM) have changed since 2000, and new classes of drug have been introduced. Our aim was to describe how drug choice at initiation and first stage of intensification have changed over this period, and to what extent prescribing was in accord with clinical guidelines, including adherence to recommendations regarding kidney function.

**Design** Repeated cross-sectional study.

**Setting** UK electronic primary care health records from the Clinical Practice Research Datalink.

**Participants** Adults initiating treatment with a drug for T2DM between January 2000 and July 2017.

**Primary and secondary outcome measures** The primary outcomes were the proportion of each class of T2DM drug prescribed for initiation and first-stage intensification in each year. We also examined drug prescribing by kidney function and country within the UK.

**Results** Of 280 241 people initiating treatment with T2DM drugs from 2000 to 2017, 73% (204 238/280 241) initiated metformin, 15% (42 288/280 241) a sulfonylurea, 5% (12 956/280 241) with metformin and sulfonylurea dual therapy and 7% (20 759/280 241) started other options. Clinicians have increasingly prescribed metformin at initiation: by 2017 this was 89% (2475/2778) of drug initiations. Among people with an estimated glomerular filtration rate of $\leq$30 mL/min/1.73 m$^2$, the most common drug at initiation was a sulfonylurea, 58% (659/1135). In 2000, sulfonylureas were the predominant drug at the first stage of drug intensification (87%, 534/615) but by 2017 this fell to 30% (355/1183) as the use of newer drug classes increased. In 2017, new prescriptions for dipeptidyl peptidase-4 inhibitors (DPP4i) and sodium/glucose cotransporter-2 inhibitors (SGLT2i) accounted for 42% (502/1183) and 22% (256/1183) of intensification drugs, respectively. Uptake of new classes differs by country with DPP4is and SGLT2is prescribed more in Northern Ireland and Wales than England or Scotland.

**Conclusions** Our findings show markedly changing prescribing patterns for T2DM between 2000 and 2017, largely consistent with clinical guidelines.

## INTRODUCTION

In the UK, the vast majority of prescribing for type 2 diabetes mellitus (T2DM) is undertaken within primary care. The aim of treatment is to reduce hyperglycaemia and morbidities associated with T2DM, such as cardiovascular

### Strengths and limitations of this study

► This study uses contemporary UK primary care data to examine how prescribing at the first stage of treatment intensification for type 2 diabetes after metformin monotherapy has changed from 2000 to 2017.

► Using long-term prescribing data has enabled us to compare people at the same stage of treatment.

► We may have included some patients with type 1 diabetes, and may have wrongly classified some people who were changing rather than intensifying treatment.

disease and microvascular complications such as chronic kidney disease (CKD) or retinopathy.[1 2] National Institute for Health and Care Excellence (NICE) and Scottish Intercollegiate Guidelines Network (SIGN) provide clinical guidance for the management of T2DM. After lifestyle changes, both NICE (NG28) and SIGN (154) recommend a series of intensification steps, adding drugs to a baseline of metformin monotherapy and only stopping metformin if there are clinical reasons to do so.[1 2] Estimates suggest that 30%–50% of people who started treatment on metformin monotherapy in the USA and Europe went on to further drug intensification.[3 4]

There are an increasing number of potential drug classes for the first stage of intensification after metformin monotherapy. Two new drug classes have recently been introduced: dipeptidyl peptidase-4 inhibitors (DPP4is; first licensed in the UK in 2007) and sodium-glucose cotransporter-2 inhibitors (SGLT2is first licensed in the UK in 2012). Guidelines have been updated to reflect these new options (figure 1).[1 2 5 6] Sulfonylureas (SU), SGLT2is, DPP4is and thiazolidinediones (TZD) are the current drug options for the first stage of drug intensification and are associated with different risk profiles and possibly specific benefits.[7 8] In light of the changing treatment guidelines, we aimed

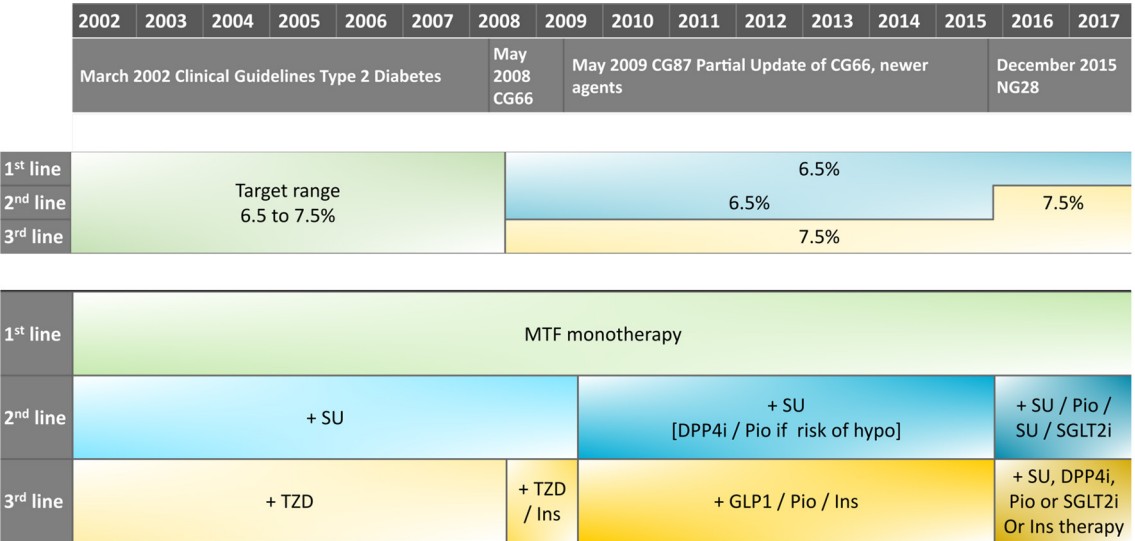

**Figure 1** Changing National Institute for Health and Care Excellence (NICE) recommendations for type 2 diabetes mellitus (T2DM) drug treatment. CG, clinical guideline; DPP4i, dipeptidyl peptidase-4 inhibitor; GLP1, glucagon-like peptide-1; Ins, insulin; MTF, metformin; NG, NICE guideline; Pio, pioglitazone; SGLT2i, sodium-glucose cotransporter-2 inhibitor; SU, sulfonylurea; TZD, thiazolidinedione.[1 2 5 6 14 21 33 34]

to describe patterns of prescribing using UK primary care data between 2000 and 2017, examining trends in prescribing at treatment initiation and at the point of first drug intensification, and to investigate the degree of concordance with guideline recommendations, in particular in relation to kidney function. In secondary analyses, we have explored whether there is variation in local practice by describing prescribing according to geographic location and clusters of general practices.

## METHODS
### Study setting
This observational study used data from the UK Clinical Practice Research Datalink (CPRD), a source of pseudonymised primary care health data which is regularly audited to ensure quality. CPRD data include demographic and lifestyle factors, records of prescriptions, clinical and test records and referrals to secondary care. The data come from primary care providers in England, Wales, Scotland and Northern Ireland and have been used extensively for clinical and pharmacoepidemiology studies, with previous validation studies suggesting that diagnoses coded in CPRD are highly reproducible from other data sources.[9 10]

### Participants
We identified all individuals aged ≥18 years who started drug treatment for T2DM between 2000 and 2017. Although the onset of T2DM is typically over the age of 40 years, the age of diagnosis is decreasing over time, and earlier onset (and longer duration) is associated with poorer patient outcomes. We therefore only excluded the very young who are substantially more likely to have type 1 diabetes mellitus (T1DM).[11] We specified that patients should be registered at a general practitioner (GP)

practice recording research quality data for a period of 12 months before starting drug treatment for diabetes to restrict the cohort to only new users of T2DM drugs.

We excluded women with a record of pregnancy (within 12 months either side of baseline prescription) as UK prescribing guidelines recommend different drug regimens for pregnant and breastfeeding women compared with other patients with T2DM.[12 13]

Codes to identify T2DM drugs were created based on British National Formulary T2DM chapters and drug codes are provided in the online supplementary file and on LSHTM compass, http://datacompass.lshtm.ac.uk/649/. We used the CPRD data released in July 2017.

### Definition of exposure, outcome and covariates
#### Drug initiation cohort
We described prescribing for two cohorts of patients. The first included individuals who received any prescriptions for their first antidiabetic drug. We identified the first T2DM drug prescribed in their patient record. Where more than one drug was prescribed on the day of initiation, the treatment was recorded as a combination therapy of the drugs prescribed.

#### First stage of drug intensification cohort
Metformin is the only drug recommended by NICE and SIGN for drug initiation, with further drugs subsequently added if greater glycaemic control is required at the first stage of intensification. Therefore, we went on to describe prescribing among patients who intensify treatment after a period of metformin monotherapy. We described the first new drug prescribed after metformin monotherapy without any time limit. We sought to do this and exclude those who switched treatment by requiring that included individuals had a further prescription for metformin within 60 days of the prescription for a new drug class.

We did not describe further prescribing for patients who switch treatment from metformin as our focus is on treatment intensification rather than switching.

## Covariates

For both cohorts we investigated how prescribing has changed over time by describing patterns for each calendar year, with year based on the day that the initiation or first intensification drug was first recorded in the patient record. Metformin is contraindicated for those with an increased risk of lactic acidosis such as those with reduced kidney function. Therefore, we also described treatment patterns for people with reduced renal function: (1) in individuals whose most recent estimated glomerular filtration rate (eGFR) was ≤30 mL/min/1.73 m² prior to drug prescription to reflect current treatment guidance, and (2) individuals with a serum creatinine higher than 130 µmol/L prior to drug prescription, to reflect guidance from 2002 that used this higher serum creatinine target.[14 15] eGFR was calculated using the last creatinine result, recorded not more than 540 days (18 months) prior to the date of treatment prescription, since we expected creatinine to be measured annually as recommended during the study period by Quality Outcomes Framework[16] and the National Diabetes Audit.[17] We calculated eGFR using the CKD-EPI equation[18] excluding the ethnicity factor as this is not entered in CPRD for a substantial proportion of individuals.[19]

To assess country-level differences, we stratified prescribing according to the location of each general practice: England, Northern Ireland, Scotland and Wales. We also described first stage of intensification prescribing according to Clinical Commissioning Group (CCG) groupings. CCGs are groups of GP practices that are responsible for commissioning local health services for patients, and may have shared management protocols or prescribing guidance. Here, GP practices are identified to be in the same CCG but there is no other identifiable information on the location of the CCG.

## Statistical analysis

To examine how drug prescribing changed over time we first described patterns using counts of drug initiations between 2000 and 2017 with total prescribing for each year as the denominator. Then, we repeated this for the first stage of intensification prescribing patterns. We described initiation prescribing in the subgroup of individuals with reduced renal function, and we provided prescribing patterns for the first stage of intensification according to country and CCG. We calculated 95% CIs for the proportions using the standard normal distribution approximation. For people intensifying treatment in 2016, we calculated the mean time between starting metformin and the second treatment, we restricted to 2016 as this was the final year with complete data, and restricting to a single year reduced differential lead time due to non-availability of newer drugs in previous years.

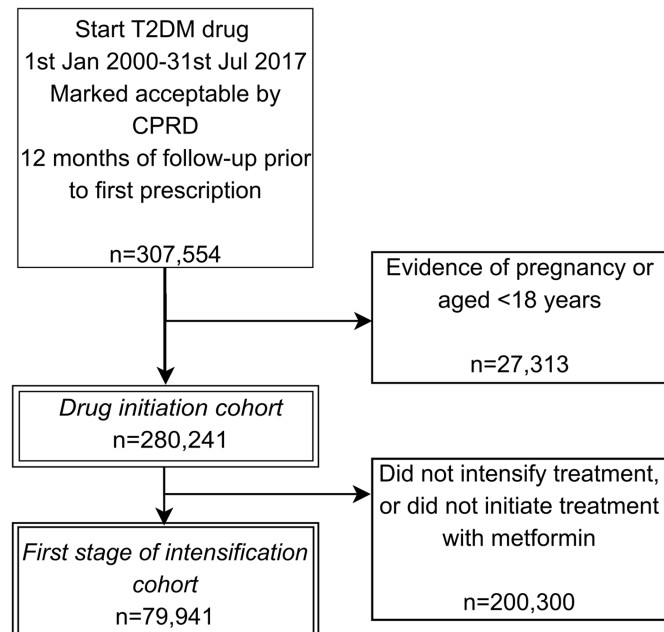

**Figure 2** Flow diagram showing the creation of the cohorts and reasons for exclusion. CPRD, Clinical Practice Research Datalink; T2DM, type 2 diabetes mellitus.

Data extraction and processing of CPRD data was completed in Stata MP (V.14). All data analyses were completed using R and R packages for reproducible research. We reported our findings according to the RECORD reporting guidelines.[20]

## Ethical and scientific approval

The research protocol was approved by the Independent Scientific Advisory Committee (ISAC) for MHRA Database Research (protocol number 16_267). The protocol was made available to reviewers for peer review. Ethical approval for observational research using CPRD GOLD with approval from ISAC has been granted by a Health Research Authority Research Ethics Committee (East Midlands-Derby, REC reference number 05/MRE04/87). This study was also approved by the London School of Hygiene and Tropical Medicine Ethics Committee (reference 11923).

## Patient and public involvement statement

Patients were not involved in the design or conduct of the study. We plan to disseminate the results through peer review publication.

## RESULTS

We identified 280 241 people initiating treatment with an antidiabetic drug between the start of 2000 and July 2017. Inclusions and exclusions are shown in figure 2. Of those initiating treatment, 204 238/280 241 (73%) initiated with metformin monotherapy, 42 288/280 241 (15%) with SU monotherapy and 12 956/280 241 (5%) with metformin and SU dual therapy. Insulin monotherapy represents 6771/280 241 (2%) of initiations for

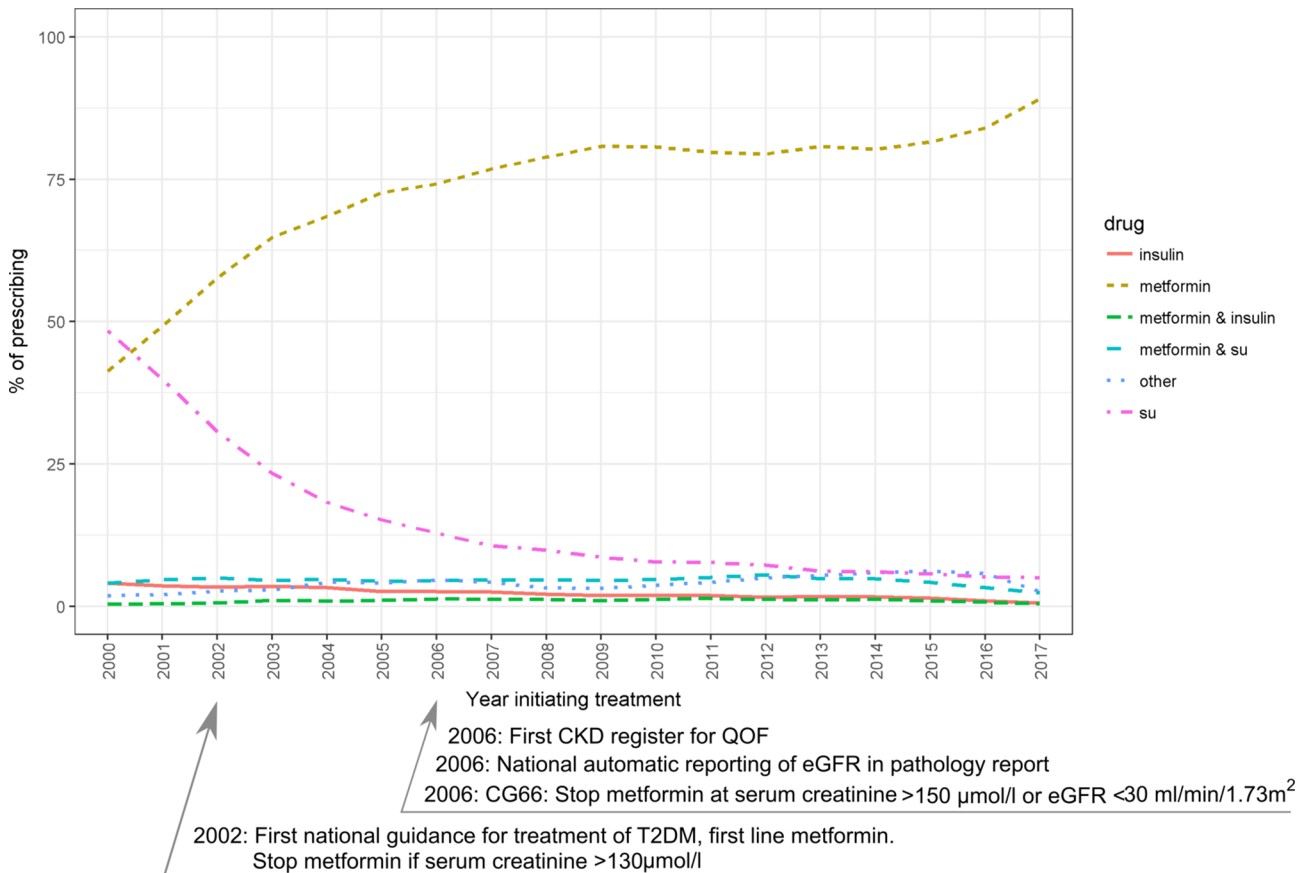

**Figure 3** Drug prescribing at T2DM drug initiation 2000–2017.[14 21 22] CG, clinical guideline; CKD, chronic kidney disease; eGFR, estimated glomerular filtration rate; QOF, Quality Outcomes Framework; SU, sulfonylurea; T2DM, type 2 diabetes mellitus.

the period and 13 988/280 241 (5%) started another drug option. Of this 5%, the most common drugs were insulin in combination with metformin (2850/13 988), TZD in combination with metformin (1405/13 988) or TZD alone (1393/13 988). A full list of combinations is provided in online supplementary table 1.

### Temporal patterns of prescribing: treatment initiation

Patterns of treatment initiation drug prescribing changed over time (figure 3 and online supplementary table 2). In 2000, GPs prescribed SU monotherapy more often than metformin monotherapy but have increasingly prescribed metformin which now accounts for 89% (95% CI 88% to 90%) of drug initiations for T2DM. A small number of people in our drug initiation cohort start treatment on insulin therapy and this declines over time. Prescribing of insulin fell from 4% in 2000 to 0.58% in 2017.

### Prescribing among people with reduced renal function

We found 145 822/280 241 (52%) people with eGFR measured in the 540 days prior to initiating drug therapy. Of these 1135/145 822 (1%) had an eGFR ≤30 mL/min/1.73 m$^2$ and 5395/145 822 (4%) had a serum creatinine ≥130 µmol/L. Among people with an eGFR ≤30 mL/min/1.73 m$^2$ the most common drug for initiating treatment was an SU at 58% (659/1135) of total prescribing from 2000 to 2016. Prescribing of metformin as the first

drug in this group fell steadily from 29% (95% CI 28% to 30%) in 2000 to 9.5% (95% CI 9% to 10%) in 2016. Since being licensed in 2007, prescriptions for DPP4is as initial therapy for this subgroup have steadily increased to 33% (95% CI 32% to 34%) in 2016. Full details of prescribing are supplied in online supplementary table 3. A comparison of initiation drug prescribing between the current and earlier guidance on renal function is presented in online supplementary figure 1.

### Temporal patterns of prescribing: first stage of drug intensification

Of the individuals who started metformin monotherapy, we identified 105 348/277 232 (38%) people who started on metformin and then received a second class of T2DM drug. Of these, 79 941/105 348 (76%) were prescribed metformin in the 60 days after the new drug prescription, indicating treatment intensification rather than switching. Among these 79 941 people, the drugs prescribed at the first stage of drug intensification have changed over the period of the study (figure 4).

In 2000, SU prescribing dominated drug choices at the first stage of intensification, accounting for 87% (95% CI 84% to 90%) of new drug intensifications. By 2017, this fell to 30% (95% CI 25% to 35%). Between 2000 and 2006, there was a rise in the use of TZD class prescribing, but after

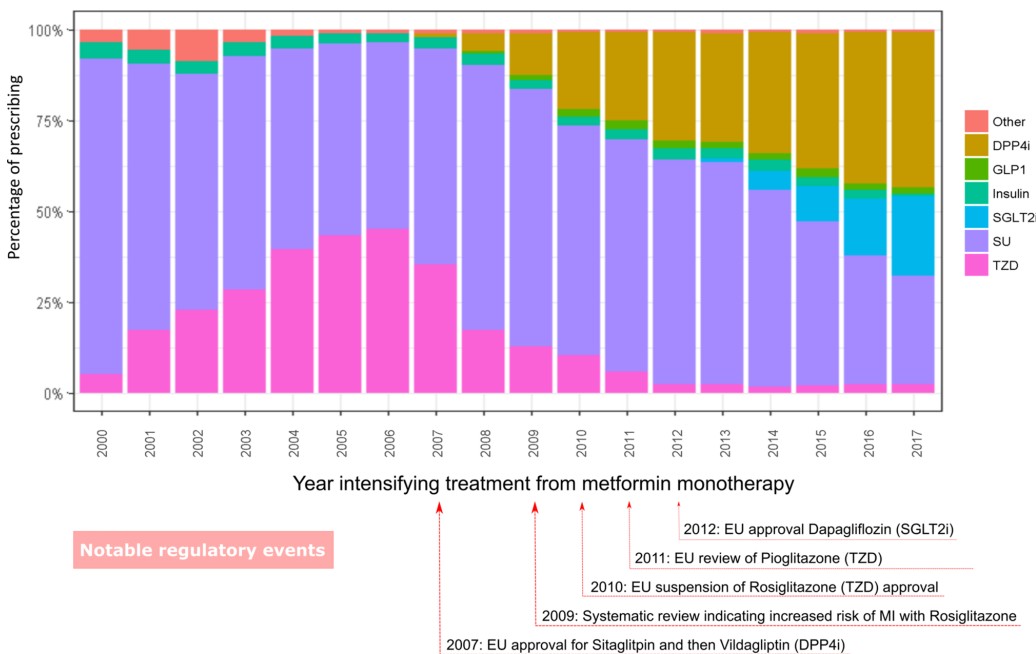

**Figure 4** First-stage intensification prescribing as a percentage of total prescribing 2000–2017.[31 35 36] DPP4i, dipeptidyl peptidase-4 inhibitor; EU, European Union; GLP1, glucagon-like peptide-1; MI, myocardial infarction; SGLT2i, sodium/glucose cotransporter-2 inhibitor; SU, sulfonylurea; TZD, thiazolidinediones.

2006, TZD use fell. In 2017, TZD prescribing accounted for only 2% (95% CI 0% to 8%) of prescribing, compared with a peak of 45% (95% CI 43% to 47%) in 2006. Prescribing of two new drug classes, DPP4is and SGLT2is increased since their introduction in 2007 and 2012, respectively. In 2017, new prescriptions for DPP4is accounted for 42% (95% CI 38% to 47%) of first stage of intensification drug choices. SGLT2i prescribing is rising, accounting for 22% (95% CI 17% to 27%) of new drug intensifications in 2017 (online supplementary table 4). Other than insulin (about 21 months) the other drugs were all started after a similar

time period following metformin monotherapy (around 3–3.7 years) (online supplementary table 5).

## REGIONAL DIFFERENCES

Prescribing practice differs between countries within the UK (figure 5, online supplementary table 6). For 2013–2017, GPs in Wales and Northern Ireland prescribed DPP4is in 45% (95% CI 42% to 48%) and 46% (95% CI 41% to 51%) of intensifications whereas in Scotland and England GPs prescribed DPP4is in just 30% (95% CI 26% to 34%) and

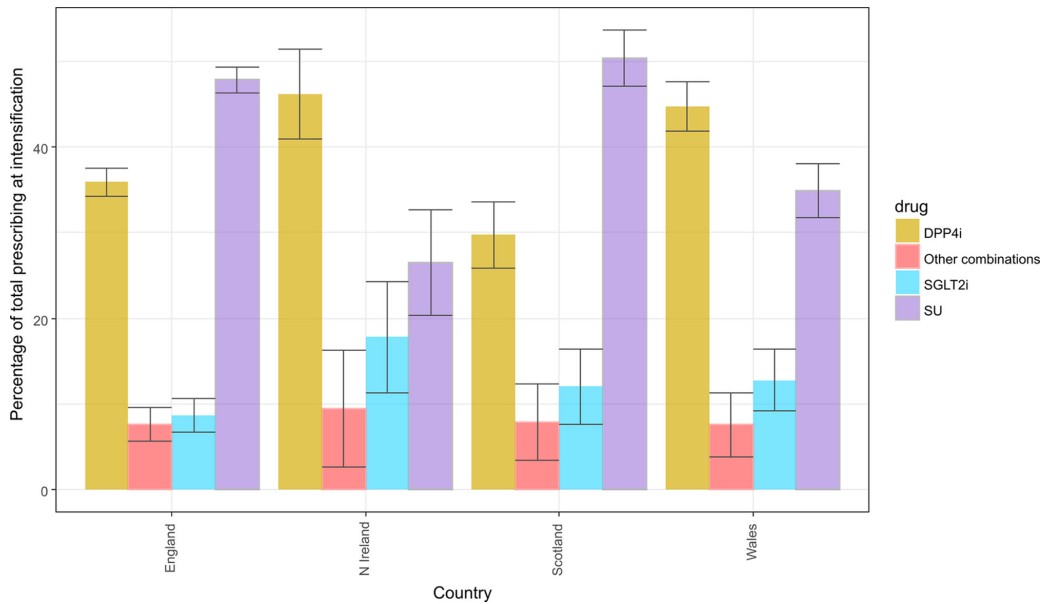

**Figure 5** Proportions of patients at first-stage intensification prescribed a DPP4i, SU, SGLT2i and other drugs, by country, 2013–2017. DPP4i, dipeptidyl peptidase-4 inhibitor; SGLT2i, sodium/glucose cotransporter-2 inhibitor; SU, sulfonylurea.

36% (95% CI 34% to 38%) of patients intensifying treatment. GPs in Northern Ireland prescribed SGLT2is in 18% (95% CI 11% to 24%) of intensifications compared with 13% (95% CI 9% to 16%) in Wales, 12% (95% CI 8% to 16%) in Scotland and 9% (95% CI 7% to 11%) in England. We also found marked heterogeneity of prescribing practice across CCG groupings (online supplementary figure 2).

## DISCUSSION

Our results show that prescribing of metformin has increased and prescribing of SUs has fallen at drug initiation for T2DM between 2000 and 2017, and shows increasing accordance with recommendations from national and international guidelines. In patients with an eGFR ≤30 mL/min/1.73 m$^2$ the most commonly prescribed initiation drug class was SUs until 2015, but since then DPP4is are more commonly prescribed. Of note, we found that approximately 1 in 10 people are prescribed metformin despite an eGFR ≤30 mL/min/1.73 m$^2$. Prescribing patterns at the first stage of drug therapy intensification have also changed, with prescribing of SUs and TZDs falling, while that of newer drug classes has risen. By 2017, the most commonly prescribed addition to metformin was a DPP4i. Prescribing practice differs by country within the UK. We identified large differences in prescribing practice between countries in the UK, with Northern Ireland and Wales prescribing both DPP4is and SGLT2is more commonly than in England or Scotland. We also show large variations in prescribing practice between CCGs.

Our large study uses data from a source of population representative primary care records from across the UK to provide great insight into real-world clinical practice from 2000 to 2017. We have attempted to improve direct comparability by developing cohorts that reflect distinct stages of the management of patients with T2DM, rather than examining total prescribing. We have been able to characterise renal function prior to drug initiation for the majority of patients to explore changing concordance with prescribing recommendations.

However, there are limitations to this analysis. We do not know if the prescribing was initiated in primary or secondary care. In the absence of wider demographic features about the CCGs such as age, socioeconomic status or ethnicity distributions we cannot explore factors that might drive variation in prescribing. For some patients, more recent eGFR measures may have been available to the prescribing GP in letters or discharge summaries from secondary care, while the result available to us from serum creatinine tests could have been measured during a previous acute illness. This misclassification may in part explain why, even in recent years, nearly 10% of patients appear to initiate treatment with metformin despite levels of renal function that should have contraindicated its use. We have not analysed drug intensification patterns for patients who did not initiate treatment with a period of metformin monotherapy although this is a small minority over recent years. We may have included a proportion of

patients with T1DM, both those who commenced treatment with insulin, and those who started on drug therapy but were later reclassified. However, people commencing insulin accounted for only 2% of drug initiations over the whole period so this is unlikely to have a substantial impact on our results. Finally, since our definition of intensification was based on receiving a further metformin prescription, we may have misclassified some patients as switching from metformin monotherapy rather than intensifying treatment. For example, we will have excluded some patients who died after intensifying treatment before receiving a further metformin prescription.

The prescribing trends we identified are in keeping with a study completed using a different source of UK primary care data that examined prescribing up to 2013.[23] International comparisons also show similar trends with falls in SU prescriptions and increases in metformin use, accounting for 68% of treatment initiations in Italy in 2012, 77% in the USA in 2016, while our estimate was 84%.[3 24]

Our work has also highlighted an increase in prescribing of DPP4is for treatment intensification, similar to findings in the UK and the USA.[23 25] The additional period to 2017 covered by our analysis shows that these trends continued, with additional growth in SGLT2i prescribing.

Our results are also consistent with data from OpenPrescribing, a website that allows access to absolute numbers of near real-time GP prescriptions.[26] OpenPrescribing shows increased prescribing of DPP4is and SGLT2is but does not distinguish prescribing at different stages of treatment as we present here.

In relation to prescribing for patients with reduced renal function, our work mirrors prescribing trends from a recent US study that described prescribing over time in people with CKD, in particular the increasing use of DPP4is.[27] Our finding that metformin continues to be prescribed for patients with T2DM and severely impaired renal function echoes work from France which found that for a cohort of people with reduced renal function prescribed metformin, the prescription was against contraindications in 49% of cases, and Italy where 15% of participants with an eGFR <30 mL/min/1.73 m$^2$ were still prescribed metformin.[28 29]

Encouragingly, we have found that prescribing at initiation of drug treatment for T2DM largely follows national guidelines and concordance has improved over time. We have highlighted that uptake of new drugs at the first stage of intensification has increased rapidly over recent years with marked regional variation suggesting factors outside of clinical indication may be important; guidance from local bodies to CCGs, drug company marketing, local secondary care practice and patient demand may all influence prescribing.[30] Growing evidence that SGLT2is may offer long-term benefits for prevention of cardiovascular disease, results not previously seen for other treatments, may also have influenced prescribing, although guidelines have not yet been altered.[20 31] Increasing use of patented drugs will drive up prescribing costs, an issue of concern as drugs for diabetes now account for

approximately 10% of the total cost of National Health Service primary care prescribing spending.[32]

In conclusion, our results showed marked changes in prescribing for T2DM since 2000 with large increases in prescribing of the new agents. There is substantial variation between regions and CCGs, despite no national guidance towards prescribing of specific agents. The factors underlying choice of drug options for the first stage of intensification are unexplained, and whether drug choice affects future clinical outcomes needs to be determined.

**Contributors** SVW, ID, LS, HSF and LT were responsible for developing the research question. SVW, LT and ID completed the data analysis and summarised the results. SW drafted the manuscript. All authors (SVW, ID, HSF, DF, AP, LS, LT) have read, commented on and approved the final manuscript.

**Funding** This study was supported by GlaxoSmithKline (GSK) through a PhD scholarship for SVW. LT is funded by a Wellcome Trust Intermediate Clinical Fellowship (101143/Z/13/Z). HSF is a full-time employee of GSK. LS was supported by a Wellcome Trust Senior Research Fellowship in Clinical Science (098504/Z/12/Z). ID is paid by an unrestricted grant from GSK.

**Competing interests** SVW is funded by a GSK PhD scholarship. HSF is an employee of and holds shares in GSK. ID is funded by, holds stock in and has consulted for GSK. LS is funded by grants from Wellcome and has received grants from MRC and NIHR, grants and personal fees from GSK, grants from BHF, grants from Diabetes UK, outside the submitted work, and is a Trustee of the British Heart Foundation. AP reports personal fees from Novo Nordisk, and personal fees from Boehringer and Lilly, outside the submitted work. DF reports personal fees from ACI Clinical for clinical trial adjudication, outside the submitted work.

**Patient consent** Not required.

**Ethics approval** Health Research Authority Research Ethics Committee.

**Provenance and peer review** Not commissioned; externally peer reviewed.

**Data sharing statement** No further data sharing is possible.

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
