## [Reviewer comments · BMJ Open]

ARTICLE DETAILS

TITLE (PROVISIONAL)	Changing use of antidiabetic drugs in the UK: Trends in prescribing 2000-2017
AUTHORS	Wilkinson, Samantha; Douglas, Ian; Stirnadel-Farrant, Heide; Fogarty, Damian; Pokrajac, Ana; Smeeth, Liam; Tomlinson, Laurie

VERSION 1 – REVIEW

REVIEWER	Shota Hamada Institute for Health Economics and Policy, Japan
REVIEW RETURNED	26-Mar-2018

GENERAL COMMENTS	The authors described the prescribing trend of antidiabetic drugs from 2000 to 2017 using the CPRD. This study shows new interesting data, in particular, on rapidly increasing use of newer drug classes in recent years. The study was well conducted and the paper is well written with detailed data. I have a few suggestions to improve the manuscript. 1) How long did the authors examine add-on drugs after 60 days of metformin monotherapy? Also, how long did it take from metformin initiation to the first intensification? 2) Older people account for a large proportion of newly diagnosed people with type 2 diabetes. Given that the individualization of HbA1c targets has been emphasized for older people, are there any differences in drug utilization, especially for intensification, between younger and older patients and between the past years and recent years? I also suggest the authors include key demographic data for the study participants over time in the manuscript or supplement.
--

REVIEWER	Annalisa Capuano Department of Experimental Medicine - University of Campania "Luigi Vanvitelli" - Naples (Italy)
REVIEW RETURNED	30-Mar-2018

GENERAL COMMENTS	This is an interesting cross-sectional study, which has investigated the prescription pattern of antidiabetic drugs for initiation and first stage intensification with the aim to provide great insight into real world clinical practice between 2000 and 2017. According to study results, clinicians have increasingly prescribed metformin at initiation, which is in line with current European guidelines, but authors also found that there was an increase in prescribing of DPP4is for treatment intensification, and falls in sulfonylurea prescribing. In my opinion, there are just few flaws that need to be addressed in a revised version.
--

	- Authors should improve some typing and grammatical errors in the entire manuscript (see, for example, the beginning of page 16 of 53). - In the discussion (“The prescribing trends we identified are in-line with 2016, whilst our estimate was 84%.(26)”), authors should compare their results also with those obtained from other European or American studies in which there was a similar drop in sulfonylurea prescription and increase DPP4is prescription (just few lines – see for example Rafaniello C et al. “Trends in the prescription of antidiabetic medications from 2009 to 2012 in a general practice of Southern Italy: a population-based study” and Hampp C et al. “Use of antidiabetic drugs in the U.S., 2003-2012” .
--	--

REVIEWER	NUR LISA ZAHARAN Department of Pharmacology, Faculty of Medicine University of Malaya, Malaysia
REVIEW RETURNED	04-Apr-2018

GENERAL COMMENTS	Thank you for the opportunity to review this interesting manuscript. This manuscript described the prescribing trends of antidiabetic drugs in the UK using the UK-CPRD database. The importance of this study is clear as the prevalence of diabetes increases and newer medications such as the SGLT-2 inhibitors are being increasingly prescribed. This will allow health policy makers to make projection of future needs. The UKCPRD database used in this study is a validated database and is considered as one of the gold standard in health administrative databases. This database provided valuable information on the prescribing at the level of primary care and has been used in many other studies to gain insight into the UK practice. Although the data used in this study covers 7% of the UK population, the quality and coverage indicate that the results could be generalized to the population. Abstract In the abstract, the authors did not mention in the objective that this study also examined the prescribing trends for patients with diabetes and concurrent reduced GFR functions. However, this part was highlighted in both the methods and results section. To improve the novelty of this manuscript, focus should shift towards the changing trends on the prescribing of the newer antidiabetic agents. Background Please provide reference for NICE and SIGN clinical guideline (Paragraph 1) Please revise the last sentence in Paragraph 1 to make it easier to understand. Perhaps include a statement on the agreement of the guideline for initiation of treatment with metformin and the intensification steps. I suggest including the expected proportion of patients who would require additional therapies from the literature. Please make it clearer in the introduction section that this study examined initiation and first stage of intensification rather than overall prescribing trends for patients with diabetes and the rationale of this approach.
---

	In the second paragraph, the addition of the new drug in the market was mentioned, and the authors mentioned that in the last paragraph “In light of the changing treatment guidelines, we aimed to describe patterns of prescribing using UK primary care data between 2000 and 2017, and to investigate the degree of concordance with guideline recommendations. It would be of interest to consider if the timing of the changing guideline (for example, calendar year 2009 for GLP and 2015 for SGLT-2- Figure 1, Figure 4) impact the prescribing rate using segmented regression analysis. As mentioned above, this study also examined the prescribing of antidiabetic agents in patients with renal impairment (eGFR<30 ml/min/1.73m²). However, there was no mention of this part in the introduction section. In addition, the rationale of examining geographical variations were not included. Methodology Study setting: Please mention previous validation of this database. Participants: Do you exclude pregnant women for drug intensification study? As this study examined prescribing trends over the years, I would suggest examining for time trend and describe if the change of prescribing rates for metformin and sulphonylureas were significantly different compared to baseline prescribing in 2000 and perhaps, to include a segmented regression analysis based on the changes in guideline (Figure 4, please see comment above) Please briefly describe the Clinical Commissioning Group (CCG) groupings for non UK readers. Consider performing further analysis (ANOVA or logistic regression) to examine the geographical variations. Results Initiation therapy: For the 5% prescribed other drugs, please include the percentage Temporal patterns of prescribing. The increase in the prescribing of metformin and sulphonylureas over the years have been described in other studies in different population. To enhance the novelty of this paper, consider focusing on the changes in the new antidiabetic agents and its relation to notable regulatory event (Figure 4) or first marketing of new drug that would add the value of this paper. Prescribing among people with reduced renal function. Consider presenting the change in the prescribing of patients with reduced renal function post 2007 in more details. Intensification of therapy: The proportion of patients who required intensification after metformin therapy (38%) seemed rather low. Do you have any other study for comparison? Discussion
--	--

	In the first paragraph, the authors mentioned the large differences in prescribing practice between country in terms of prescribing. I am of the opinion this statement should be supported with statistical significance (see comments in methodology). Please revise last sentence Paragraph 3 as it is difficult to comprehend. Consider including comparison in trends of prescribing with other countries, at least in the European region (Paragraph 4) Conclusion - The conclusion is well supported by the study but quite vague. Consider revising to enhance the impact of this paper.
--	---

VERSION 1 – AUTHOR RESPONSE

No	Comments by Editor	Response to Editor	Location of change in the revised manuscript
1	Please revise the ‘Strengths and limitations’ section of your manuscript (after the abstract). This section should relate specifically to the methods of the study.	We thank all the reviewers for taking the time to review our manuscript, and for the many useful comments and suggestions. Below we have addressed each point. We have now updated the Strengths and Limitations sections to reflect only the study methods.	Page 4, lines 2-7
No.	Comments by Reviewer 1	Response to Reviewer 1	Location of change in the revised manuscript
	The authors described the prescribing trend of antidiabetic drugs from 2000 to 2017 using the CPRD. This study shows new interesting data, in particular, on rapidly increasing use of newer drug classes in recent years. The study was well conducted and the paper is well written with detailed data. I have a few suggestions to improve the manuscript.	We very much appreciate your kind review of our manuscript. We have responded each of the points raised below.	N/A
1	How long did the authors examine add-on drugs after 60 days of metformin monotherapy? Also, how long did it take from metformin initiation to the first	To clarify, participants could start another drug at any time after starting metformin. We required that they received another prescription for metformin after the alternate intensification drug in order to confirm that they were intensifying treatment	Page 7, line 20-21 Page 9, line 12-15 Page 13, line 3-5 New table:

	intensification?	rather than changing drug therapy. We have added clarification to the methods. The mean and 95% CI for the time between starting metformin and the first intensification drug are shown in the table below, both over the whole time period and just for 2016. We have added this information to supplementary table 5 and clarified this section of the methods on page 9, and results on page 13.	Supplementary table 5
2	Older people account for a large proportion of newly diagnosed people with type 2 diabetes. Given that the individualization of HbA1c targets has been emphasized for older people, are there any differences in drug utilization, especially for intensification, between younger and older patients and between the past years and recent years? I also suggest the authors include key demographic data for the study participants over time in the manuscript or supplement.	We agree that a full exploration of drug utilisation is a very interesting extension to this work, and are currently conducting a follow on study that addresses differences in drug prescription according to both demographic and clinical factors. We aim to publish the results of this work as soon as possible.	N/A
No.	Comments by Reviewer 2	Response to Reviewer 2	Location of change in the revised manuscript
	This is an interesting cross-sectional study, which has investigated the prescription pattern of antidiabetic drugs for initiation and first stage intensification with the aim to provide great insight into real world clinical practice between 2000 and 2017. According to study results, clinicians have increasingly prescribed metformin at initiation, which is in line with current European guidelines, but authors also found that there was an increase in prescribing of DPP4is for treatment intensification, and falls in sulfonylurea prescribing. In my opinion, there are just few flaws that need to be	Thank you for your very valuable comments, they are greatly appreciated. We have addressed each of your points below.	N/A

	addressed in a revised version.		
1	Authors should improve some typing and grammatical errors in the entire manuscript (see, for example, the beginning of page 16 of 53).	Thank you for highlighting this, we have edited the manuscript in the sections given and throughout.	Page 8, line 10-11 Page 10, line 16 Page 14, line 13 Page 15, line 14 Page 16, line 5
2	In the discussion (“The prescribing trends we identified are in-line with 2016, whilst our estimate was 84%.(26)”), authors should compare their results also with those obtained from other European or American studies in which there was a similar drop in sulfonylurea prescription and increase DPP4is prescription (just few lines – see for example Rafaniello C et al. “Trends in the prescription of antidiabetic medications from 2009 to 2012 in a general practice of Southern Italy: a population-based study” and Hampp C et al. “Use of antidiabetic drugs in the U.S., 2003-2012”.	Thank you very much for highlighting these useful studies. We have added some lines in the discussion to include these studies as suggested.	Page 15, line 16-20 Page 16, line 1-3
No.	Comments by Reviewer 3	Response	Location of change in the revised manuscript
	This manuscript described the prescribing trends of antidiabetic drugs in the UK using the UK-CPRD database. The importance of this study is clear as the prevalence of diabetes increases and newer medications such as the SGLT-2 inhibitors are being increasingly prescribed. This will allow health policy makers to make projection of future needs. The UKCPRD database used in this study is a validated database and is considered as one of the gold standard in health administrative databases. This database provided valuable information	Thank you very much for your detailed and thoughtful comments. We have tried to address each suggestion below.	N/A

	on the prescribing at the level of primary care and has been used in many other studies to gain insight into the UK practice. Although the data used in this study covers 7% of the UK population, the quality and coverage indicate that the results could be generalized to the population.		
1	In the abstract, the authors did not mention in the objective that this study also examined the prescribing trends for patients with diabetes and concurrent reduced GFR functions. However, this part was highlighted in both the methods and results section.	Thank you for highlighting this, we have updated the abstract to highlight that the aspect of the guidelines that we are interested in is kidney function	Page 2, line 5-6
2	Please provide reference for NICE and SIGN clinical guideline (Paragraph 1)	Thank you, we have added this.	Page 5, line 9
3	Please revise the last sentence in Paragraph 1 to make it easier to understand.	We have revised the sentence.	Page 5, line 8-9
4	Perhaps include a statement on the agreement of the guideline for initiation of treatment with metformin and the intensification steps.	Thank you, we have now made this more clear on page 5.	Page 5, line 20-22
5	I suggest including the expected proportion of patients who would require additional therapies from the literature.	Thank you for this suggestion, we have added a reference and few lines of explanation.	Page 5, line 9-11
6	Please make it clearer in the introduction section that this study examined initiation and first stage of intensification rather than overall prescribing trends for patients with diabetes and the rationale of this approach	Thanks you, we have added the following statement: ...examining trends in prescribing at treatment initiation and at the point of first drug intensification, and to investigate the degree of concordance with guideline recommendations, in particular in relation to kidney function.	Page 5, line 20-22 Page 6, line 1-2
7	In the second paragraph, the addition of the new drug in the market was mentioned, and the authors mentioned that in the last paragraph "In light of the changing treatment guidelines, we aimed to describe patterns of prescribing using UK primary care data between 2000 and 2017, and to investigate the degree of concordance with	Thanks for this suggestion, we agree it would be of interest to examine directly how changing guidelines impact prescribing. However, we imagine that a range of factors are effecting current prescribing in this practice, in particular the availability and growing evidence base for new drugs. Analyses to examine the impact of various factors on prescribing would be taken forward in a follow on study, but are not in the scope of the current work.	N/A

	guideline recommendations. It would be of interest to consider if the timing of the changing guideline (for example, calendar year 2009 for GLP and 2015 for SGLT-2- Figure 1, Figure 4) impact the prescribing rate using segmented regression analysis.		
8	As mentioned above, this study also examined the prescribing of antidiabetic agents in patients with renal impairment (eGFR<30 ml/min/1.73m2). However, there was no mention of this part in the introduction section.	Thank you for highlighting this, we have added a further clarification to the abstract and introduction sections.	Page 2, line 5-6 Page 5, line 20-22
9	In addition, the rationale of examining geographical variations were not included.	We have added a sentence at the end of the introduction to highlight this.	Page 6, line 1-2
10	Methodology Study setting: Please mention previous validation of this database.	Thank you for suggesting this. We have added a new reference and line of explanation..	Page 6, line 16-17
11	Participants: Do you exclude pregnant women for drug intensification study?	Each investigation cohort was created from a base cohort described under participants, page 7 line 4-6. Here we explain the rationale for excluding women with a recent history of pregnancy as guidelines recommend different management. Therefore, pregnant women were excluded for all aspects of the study.	N/A
12	As this study examined prescribing trends over the years, I would suggest examining for time trend and describe if the change of prescribing rates for metformin and sulphonylureas were significantly different compared to baseline prescribing in 2000 and perhaps, to include a segmented regression analysis based on the changes in guideline (Figure 4, please see comment above)	Thank you. As discussed above our aim was to provide broad descriptive analyses with 95% confidence intervals to provide measures of precision. To reflect the most recent picture we have also focused on trends in 2016.	N/A
13	Please briefly describe the Clinical Commissioning Group (CCG) groupings for non UK readers.	Thank you very much for this suggestion, we have added some information for readers.	Page 9, lines 1-3
14	Consider performing further analysis (ANOVA or logistic	To help understand whether our findings are likely to be explained by	N/A

	regression) to examine the geographical variations.	random error we have provided 95% confidence intervals on Figure 5. We have not undertaken multivariable analysis as lack of detailed information about regions such as age or ethnicity distributions limit the strength of conclusions we can draw about differences in regional prescribing.	
15	Initiation therapy: For the 5% prescribed other drugs, please include the percentage	This is provided on page 10, 13,988/280,241 (5%) started another drug option.” A full breakdown of other drugs is supplied in the Supplementary Table 1.	Page 10, line 16
16	Temporal patterns of prescribing. The increase in the prescribing of metformin and sulphonylureas over the years have been described in other studies in different population. To enhance the novelty of this paper, consider focusing on the changes in the new antidiabetic agents and its relation to notable regulatory event (Figure 4) or first marketing of new drug that would add the value of this paper.	Thank you. We agree that our findings are consistent with other studies, though we feel that our design to compare people at the same stage of treatment is novel and useful. We also agree that it is the findings in relation to the newest class of drugs that are the most important and interesting. We feel that substantial changes in prescribing of these over time are clear from the data provided and that further statistical testing would not be clinically meaningful.	N/A
17	Prescribing among people with reduced renal function. Consider presenting the change in the prescribing of patients with reduced renal function post 2007 in more details.	Thank you: we are not certain what additional details you would like to see. We have currently placed the three images that we feel are the most important in the main paper but are happy to consider the views of the editorial team if they would like more figures in the main paper.	N/A
18	Intensification of therapy: The proportion of patients who required intensification after metformin therapy (38%) seemed rather low. Do you have any other study for comparison?	Thank you for highlighting this, we have added a further reference that shows a snapshot of prescribing across Europe in 2012 and amended line 9 page 5 in keeping with this. The study indicates that 32-46% of people starting treatment go on to intensify treatment; our estimate therefore falls in this range.	Page 5, line 8-11
19	In the first paragraph, the authors mentioned the large differences in prescribing practice between country in terms of prescribing. I am of the opinion this statement should be supported with statistical significance (see comments in methodology).	Thank you, please see our thoughts above.	N/A

20	Please revise last sentence Paragraph 3 as it is difficult to comprehend.	Thank you, we have amended the sentence: Finally, we may have misclassified some patients at drug intensification, and will not have included those who died after intensifying treatment before receiving a further metformin prescription. To now read: Finally, since our definition of intensification was based on receiving a further metformin prescription, we may have misclassified some patients as switching from metformin monotherapy rather than intensifying treatment. For example, we will have excluded patients who died after intensifying treatment before receiving a further metformin prescription.	Page 15, lines 11-15
21	Consider including comparison in trends of prescribing with other countries, at least in the European region (Paragraph 4)	Thank you, see our response to reviewer 2, No 2 above – we have added further references to international comparisons.	Page 15, line 16-20 Page 16, line 1-3
22	Conclusion - The conclusion is well supported by the study but quite vague. Consider revising to enhance the impact of this paper.	Thank you, we have amended the concluding lines to highlight the most important findings	Page 17, line 5-9

VERSION 2 – REVIEW

REVIEWER	Shota Hamada Research Department, Institute for Health Economics and Policy, Tokyo, Japan
REVIEW RETURNED	20-May-2018

GENERAL COMMENTS	The authors have addressed my comments. I think the paper has been sufficiently improved.
---

REVIEWER	Nur Lisa Zaharan University of Malaya, Malaysia
REVIEW RETURNED	04-Jun-2018

GENERAL COMMENTS	Thank you for addressing most of the issues raised in the first review. The revision reads well and more up-to-date. The objectives have been revised and the presentation of results were clearer. I have no further comments.
--